# Forecasting the magnitude of the largest expected earthquake

Robert Shcherbakov [1,2], Jiancang Zhuang [3], Gert Zöller [4] & Yosihiko Ogata[3]

The majority of earthquakes occur unexpectedly and can trigger subsequent sequences of events that can culminate in more powerful earthquakes. This self-exciting nature of seismicity generates complex clustering of earthquakes in space and time. Therefore, the problem of constraining the magnitude of the largest expected earthquake during a future time interval is of critical importance in mitigating earthquake hazard. We address this problem by developing a methodology to compute the probabilities for such extreme earthquakes to be above certain magnitudes. We combine the Bayesian methods with the extreme value theory and assume that the occurrence of earthquakes can be described by the Epidemic Type Aftershock Sequence process. We analyze in detail the application of this methodology to the 2016 Kumamoto, Japan, earthquake sequence. We are able to estimate retrospectively the probabilities of having large subsequent earthquakes during several stages of the evolution of this sequence.

---

[1] Department of Earth Sciences, University of Western Ontario, London, ON N6A 5B7, Canada. [2] Department of Physics and Astronomy, University of Western Ontario, London, ON N6A 3K7, Canada. [3] Institute of Statistical Mathematics, 10-3 Midori-Cho, Tachikawa-Shi, Tokyo 190-8562, Japan. [4] Institute of Mathematics, University of Potsdam, 14476 Potsdam-Golm, Germany. Correspondence and requests for materials should be addressed to R.S. (email: rshcherb@uwo.ca)

Bayesian methods offer indispensable tools in studies of natural hazards, particularly, earthquakes[1–3]. They can provide a suite of approaches to analyze statistical aspects of natural seismicity, where the occurrence of earthquakes is a single realization of an underlying stochastic process. They also can be used to provide inferences about future evolution of seismicity. Specifically, one can compute predictive distributions by studying past seismicity to provide probabilistic constraints, for example, on the occurrence of subsequent large earthquakes[2,4]. The occurrence of large magnitude earthquakes is a direct manifestation of multi-scale physical processes operating in the crust and upper mantle[5–8]. However, the comprehensive understanding of the mechanisms and underlying stochastic dynamics that control the occurrence of earthquakes remains one of the challenging tasks in earthquake seismology. Therefore, the problem of formulating a reliable and operational scheme for the estimation of the magnitudes of the extreme earthquakes to occur is of critical importance.

Earthquakes form clusters in space and in time. This clustering is a result of several physical mechanisms operating in the seismogenic crust. One of them is the triggering by preceding earthquakes that can lead to a cascade of events with a complicated branching structure[9]. To account for such a clustering, the Epidemic Type Aftershock Sequence (ETAS) model offers a realistic and quantifiable approximation[10–12]. Specifically, it can model the earthquake rate before and after strong earthquakes. This, in turn, allows to quantify the increased earthquake hazard after a mainshock by incorporating the triggering ability of foreshocks, a mainshock, and subsequent aftershocks. It also can be used for short-term forecasting of large earthquakes by studying past seismicity[12–16].

The problem of estimating the largest expected magnitude of an earthquake during a future time interval has been considered in the past[17,18]. In addition, it was also addressed in case of aftershocks[19,20]. A closely related problem is the estimation of the absolute maximum magnitude of an earthquake for a given seismogenic zone[1,21–23]. In this regard, the Bayesian approach combined with extreme value statistics was used to analyze this problem by assuming that the occurrence of earthquakes can be approximated by a homogeneous Poisson process[2,24]. The Bayesian methods were also used to constrain the magnitudes of the largest expected aftershocks, where the earthquake rate was modeled by the Omori–Utsu law and the events were assumed to follow a non-homogeneous Poisson process[4,25]. However, those approaches do not fully incorporate the complicated triggering structure of earthquake sequences, which are important for short-term earthquake forecasting.

A notable exception is the application of a spatio-temporal version of the ETAS model to the retrospective forecasting of early seismicity associated with the 2016 sequence of several large earthquakes in Amatrice, Italy[14]. In that work, the MCMC approach was used to sample the posterior distribution of the ETAS parameters, in order to estimate the rates of seismicity during short forecasting time intervals. However, the parameters of the ETAS model were subjected to certain constraints in order to facilitate the sampling.

The expected large aftershocks can significantly increase the earthquake hazard after the occurrence of a mainshock[19,26]. Several approaches were proposed to address this problem[16,27,28]. In this context, the issue of the catalog early incompleteness was also incorporated in a number of studies[20,29–31]. In the majority of these studies, the point estimators for the model parameters were used to compute the associated rates or probabilities. As a result, the incorporation of the uncertainties of the model parameters was not fully addressed.

In earthquake clustering, foreshocks, which precede large earthquakes, are less consistent than the occurrence of aftershocks[9,32–34]. As a result, the retrospective analysis is typically performed by stacking many foreshock sequences associated with particular regions[33,35–40]. Therefore, the incorporation of the seismicity prior to the occurrence of large mainshocks is important for short-term earthquake forecasting approaches.

In this work, we developed a framework to constrain the magnitudes of the largest expected earthquakes to occur during a future time interval by analyzing past seismicity. To accomplish this, we combined the Bayesian analysis with extreme value statistics to compute the Bayesian predictive distribution for the magnitude of the largest expected event to exceed a certain value in the near future. In the analysis, we assume that the earthquake occurrence rate can be modeled by the ETAS process, where each earthquake is capable of triggering subsequent events[10,11]. We also assume that the distribution of earthquake magnitudes can be approximated by the left-truncated exponential distribution[41]. To model the uncertainties of the model parameters, we employed the Markov Chain Monte Carlo (MCMC) method to sample the posterior distribution of the model parameters and to use the generated chain of the parameters to simulate forward in time an ensemble of the ETAS processes. In addition, we showed that the extreme value distribution for the magnitudes of the largest events in the ETAS model deviates from the Gumbel distribution. To illustrate our approach, we analyzed one recent prominent sequence, the 2016 Kumamoto, Japan, earthquake sequence, where we were able to compute the probabilities of having the largest expected events above certain magnitudes to occur during several stages of the sequence. We also applied the analysis to several past major aftershock sequences and compared the obtained probabilities with the ones computed using several methods based on the Omori–Utsu approximation of the aftershock rate. As a main result of this work, we developed an inference procedure to estimate the probabilities of having largest expected events during an earthquake sequence described by the ETAS process. The suggested approach can be implemented in current or future operational earthquake forecasting schemes, where the constraints on the magnitudes of future large earthquakes are taken into account.

## Results

**Bayesian predictive distribution.** The occurrence of earthquakes can be modeled as a stochastic marked point process in time[42]. In this description, the earthquakes are characterized by their occurrence times $t_i$ and the earthquake magnitudes $m_i$ represent marks. The seismicity during a specified time interval can be described by an ordered set $S = \{(t_i, m_i)\} : i = 1, \ldots$. A reasonable approximation is to assume that earthquake magnitudes can be described by a parametric distribution function $F_\theta(m)$, where $\theta$ is a set of parameters. In addition, the occurrence of earthquakes is controlled by a conditional time-dependent rate, $\lambda_\omega(t | \mathcal{H}_t)$, where $\omega$ is a set of model parameters and $\mathcal{H}_t$ represents the history of past seismicity up to time $t$.

In the analysis that follows, we consider past earthquakes during a training time interval $[T_0, T_e]$. In order to take into account the effect of early events, we use the earthquakes in the first part of this interval $[T_0, T_s]$ to estimate properly the conditional earthquake rate during the target time interval $[T_s, T_e]$, for which the model parameters $\theta$ and $\omega$ can be inferred. We also consider a forecasting time interval $[T_e, T_e + \Delta T]$ during which we compute the probabilities of having extreme events above a certain magnitude (Fig. 1).

The power of the Bayesian approach is in its ability to provide predictions for an unobserved quantity of interest[3,43]. It goes beyond the traditional methods of the parameter estimation and the computation of a future outcome based on a prescribed

**Fig. 1** Schematic illustration of an earthquake sequence in time. The star symbols indicate individual earthquakes with the symbol sizes proportional to earthquake magnitudes. The earthquake times and magnitudes, $S = \{(t_i, m_i)\}$:$i = 1, ...,$ are used to sample the model parameters during the training time interval $[T_0, T_e]$ (with dark cyan color indicating significant events). Darker symbol colors separate the events in the training time interval from the events in the forecasting time interval (lighter symbol colors). The probabilities to have strong earthquakes are estimated during the forecasting time interval $[T_e, T_e + \Delta T]$ (the symbol with light cyan color indicates the largest expected earthquake)

distribution. Within the Bayesian framework, it is possible to formulate a problem of constraining the magnitudes of extreme earthquakes to occur in a future time interval $[T_e, T_e + \Delta T]$. For this problem, assuming that the occurrence of earthquakes is described by a parametric stochastic point process, one is interested in computing the extreme value distribution, $P_{EV}(m_{ex} > m|\theta, \omega, \Delta T)$. The extreme value distribution allows to estimate the probability to have the maximum event larger than a certain magnitude $m$ during the future time interval $[T_e, T_e + \Delta T]$ given the specific values of the model parameters $\theta^{\star}$ and $\omega^{\star}$. However, the precise value of the model parameters is not known for real seismicity. To overcome this difficulty, one can analyze past seismicity during the training time interval $[T_0, T_e]$ in order to compute the posterior distribution $p(\theta, \omega|S)$ for the variability of the model parameters. Therefore, using this inferred information on the model parameters one can assess the plausibility for the distribution of extreme earthquakes. This can be accomplished by computing the Bayesian predictive distribution for the largest expected event $m_{ex}$ to be larger than a certain value $m$[2,4,25]:

$$P_B(m_{ex} > m|S, \Delta T) = \int_{\Omega} \int_{\Theta} P_{EV}(m_{ex} > m|\theta, \omega, \Delta T) p(\theta, \omega|S) d\theta d\omega,$$
(1)

where $\Theta$ and $\Omega$ define the multidimensional domains of frequency-magnitude distribution and earthquake rate parameters. The predictive distribution, Eq. (1), is computed by marginalizing the model parameters and effectively incorporates into the analysis the uncertainties associated with them.

Similarly, it is possible to consider the problem of estimating a time interval $\Delta T$ to the next largest expected earthquake to be above $m_{ex}$, where the time interval is a random variable[2,4]. In this case, the probability of the next largest expected earthquake during this time interval is

$$P_B(\Delta T \leq t|S, m_{ex}) = \int_{\Omega} \int_{\Theta} P_{EV}(\Delta T \leq t|\theta, \omega, m_{ex}) p(\theta, \omega|S) d\theta d\omega.$$
(2)

By fixing a given probability level, one can estimate the interarrival time $t = \Delta T$ to the next largest earthquake with the magnitude greater than $m_{ex}$.

As stated above, the occurrence of earthquakes can be modeled as a marked point process. From this point process one can extract a sequence of exceedance events, where each event has a magnitude above a certain threshold $m_{ex}$. This threshold model is characterized by exceedance arrival times and exceedances[44,45]. The statistics of extreme events can be obtained from the analysis of exceedances over a predefined high threshold. In this framework, Eq. (2) gives the distribution of interarrival time intervals between such exceedance events.

**Extreme value distribution**. A specific functional form for the extreme value distribution, $P_{EV}(m_{ex} > m|\theta, \omega, \Delta T)$, depends on the underlying distribution function for magnitudes, $F_\theta(m)$, and whether they are independent and identically distributed (i.i.d.) events. In the case of i.i.d. events and the exponential distribution for $F_\theta(m)$, this is a well-known Gumbel distribution, which is a special type of the general extreme value (GEV) distribution[44]. The standard Gumbel distribution is defined as follows $G_I(z) \equiv \Pr\left\{ \max_j \left( X_j \leq z \right) \right\} = \exp(-e^{-z})$ and plays an important role in extreme value statistics. It provides the distribution of maxima for physical quantities that can be described by the exponential family of distributions[44,45]. However, when considering marked point processes, the extreme value distribution for extreme events may deviate from the GEV distribution due to stochastic nature of the process. In this case, one can construct the estimate of the extreme value distribution by stochastic simulation of the underlying marked point process and extracting the maximum magnitude from each simulated sequence (see the "Methods" section).

**Posterior distribution function**. In statistical parametric modeling using the Bayesian framework, the observed data can be used to constrain the variability of the model parameters by computing the posterior distribution. The prior information on these parameters is provided by using, for example, expert opinion or past studies. For the current study, considering the information for the magnitudes and times of $N_e$ earthquakes $S_{N_e}$ observed during the time interval $[T_s, T_e]$, the posterior distribution function, $p(\theta, \omega|S_{N_e})$, updates any prior information on the model parameters $\theta$ and $\omega$ by incorporating the observational data $S_{N_e}$ through the likelihood and has the following form:

$$p(\theta, \omega|S_{N_e}) \propto L\left(S_{N_e}|\theta, \omega\right) \pi(\theta, \omega),$$
(3)

where $\pi(\theta, \omega)$ is the prior knowledge for the model parameters.

The likelihood function $L\left(S_{N_e}|\theta, \omega\right)$ for a marked point process driven by the time-dependent conditional rate $\lambda_\omega(t|\mathcal{H}_t)$ with event magnitudes distributed according to $F_\theta(m)$, can be written[42]

$$L\left(S_{N_e}|\theta, \omega\right) = e^{-\Lambda_\omega(T)} \prod_{i=1}^{N_e} \lambda_\omega(t_i|\mathcal{H}_{t_i}) \prod_{i=1}^{N_e} f_\theta(m_i),$$
(4)

where $f_\theta(m) = \frac{dF_\theta(m)}{dm}$ is the probability density function and $\Lambda_\omega(T) = \int_{T_s}^{T_e} \lambda_\omega(t|\mathcal{H}_t) dt$ is the productivity of the process during the time interval $[T_s, T_e]$. When specifying the likelihood function, Eq. (4), we explicitly assume that event magnitudes are i.i.d. random variables.

**Parametric earthquake model**. To proceed with the analysis, we need to specify parametric models for the frequency-magnitude distribution of earthquake marks and for the occurrence rate of earthquakes conditioned on the past seismicity. The choice of the models is typically based on some physical grounds or past empirical studies. In this analysis, we assume that the earthquake magnitudes follow a left-truncated exponential distribution:

$$f_\theta(m) = \beta \exp[-\beta(m - m_c)], \tag{5}$$

$$F_\theta(m) = 1 - \exp[-\beta(m - m_c)], \qquad \text{for} \quad m \geq m_c, \tag{6}$$

where $\theta = \{\beta\}$. The parameter $\beta$ is related to the $b$-value of the Gutenberg–Richter scaling relation, $\beta = \ln(10)b$[41]. $m_c$ is a given lower magnitude cutoff set above the catalog completeness level.

For the conditional rate (intensity) for the occurrence of earthquakes $\lambda_\omega(t|\mathcal{H}_t)$ at a given time $t$, we assume that it can be described by the ETAS model with a given set of parameters $\omega = \{\mu, K, c, p, \alpha\}$[10,12]:

$$\lambda_\omega(t|\mathcal{H}_t) = \mu + K \sum_{i:t_i < t}^{N_t} \frac{e^{\alpha(m_i - m_0)}}{\left(\frac{t - t_i}{c} + 1\right)^p}, \tag{7}$$

where $m_0$ is a reference magnitude. The summation is performed over the history, $\mathcal{H}_t$, of past events up to time $t$ during the time interval $[T_0, t[$. $N_t$ is the number of earthquakes in the interval $[T_0, t[$ above the lower magnitude cutoff $m_c$. We also assume that $m_0 = m_c$. In the ETAS model, it is postulated that some of the earthquakes occur randomly with a background rate $\mu$. These earthquakes are typically associated with tectonic driving and are modeled by a Poisson process. In addition, each earthquake is capable of triggering subsequent earthquakes. This is reflected in the summation term in Eq. (7). As a result, the combined conditional earthquake rate at any given time, $\lambda_\omega(t|\mathcal{H}_t)$, is a superposition of contributions from the background seismicity and the triggering by preceding earthquakes defined by its history $\mathcal{H}_t$.

**The 2016 Kumamoto earthquake sequence**. To illustrate the applicability of the above formulated method, we analyzed in detail, the 2016 Kumamoto, Japan, earthquake sequence[46]. The sequence started on 14 April 2016 (12:26 UTC), when a large magnitude M6.5 foreshock occurred (Fig. 2). It is followed by the aftershock sequence that resulted in the occurrence of a magnitude M7.3 mainshock on 16 April 2016 (16:25 UTC on April 15). The largest aftershock in the sequence had magnitude M5.9, which occurred in the first day after the mainshock. In this analysis, we considered the sequence of events starting from the first foreshock of magnitude M6.5. We assigned $T_0 = 0$ to the time of this event and all subsequent event times in days are measured from this date.

To calculate the Bayesian predictive distributions and the probabilities for the largest expected earthquakes, we first generated the Markov chains for all the model parameters $\{\theta, \omega\} = \{\beta, \mu, K, c, p, \alpha\}$ by specifying the training time interval from $T_0$ to $T_e$ and extracting the earthquakes in this interval above a certain threshold $m_c$. The Markov chains were simulated using the Metropolis-within-Gibbs algorithm as described in the "Methods" section. Finally, for each set of the model parameters from the generated MCMC chain, we simulated the ETAS process forward in time using the well-established thinning algorithm[42] and extracted the maximum magnitude event. The distribution of these maxima converges to the Bayesian predictive distribution (see the "Methods" section). For the prior distribution, $\pi(\theta, \omega)$, of the model parameters, we used a Gamma distribution (specific values for the mean and variance of the priors for each model parameter are provided in Table 3). We used the point estimates

of the ETAS parameters as the mean values for the prior distributions. We also considered a truncated Normal distribution as a prior distribution. Both these distributions produced statistically similar results. As these distributions are used to constrain any prior knowledge on the model parameters, we assumed that this information can be represented by a bell-shaped distribution with a well-defined mean value and variance. The prior distribution plays an important role in the Bayesian analysis and we explore the sensitivity of the obtained results due to the variation of the prior parameters later in the analysis.

For the proposal distribution $J(x|\tilde{x})$, we used a truncated Normal distribution with the support defined as the positive real axis $x \in [0, \infty]$ to assume the positive values for the model parameters (the variances are provided in Table 3). The choice of the proposal distribution depends on how well it can approximate the posterior distribution. It also plays a critical role for the convergence of the Markov chains. Typically, standard statistical distributions are used. For each combination of the training and forecasting time intervals, we simulated total of 200,000 samples and discarded 100,000 samples as burn-in. One particular example of the Markov chains for the ETAS model parameters is given in Supplementary Fig. 1. For the MCMC sampling of the ETAS parameters we used only events during the training time interval $[T_0, T_e]$.

For the first sequence, we considered the following training time interval $T_0 = 0$, $T_s = 0.05$, and $T_e = 2.16$ days, which included the time of the occurrence of the M7.3 mainshock (Fig. 2). This includes 1 day of aftershocks that occurred after the M7.3 mainshock. We used the times and magnitudes of the earthquakes above magnitude $m_c \geq 3.3$ during this training period to compute the Bayesian predictive distribution and estimate the probabilities for the largest expected earthquakes above magnitudes $m = 5.8$ and 6.3 to occur in the next $\Delta T = 5$, 10, and 15 days. The corresponding distributions and probabilities are shown in Fig. 3. The probabilities display a gradual increase with increasing forecasting time interval $\Delta T$. The computed probability of $P_B(m_{ex} > 5.8) = 0.63$ to have an aftershock above magnitude 5.8 during the next $\Delta T = 5$ days can be appreciated by examining retrospectively the actual earthquakes that occurred during this forecasting time interval. In the sequence 2 strong aftershocks with magnitudes 5.8 and 5.5 happened during this forecasting time interval (Fig. 2).

For this sequence, we also analyzed the distribution of the interarrival time intervals $\Delta T$ to the next largest expected earthquake to be above a certain magnitude $m_{ex}$ (Fig. 4). We computed the distribution of interarrival times to the next extreme earthquake above several magnitude thresholds $m_{ex} = 5.8$, 6.3, and 6.5. For the fixed probability levels of 5%, 10%, and 20% the estimated interarrival times are given in the legend of Fig. 4.

In addition to the sequence analyzed above, we also considered the training time interval from $T_0 = 0$ to $T_e = 1.16$ days with $T_s = 0.03$ days, which ended right before the occurrence of the M7.3 mainshock. We used the times and magnitudes of the earthquakes above magnitude $m_c \geq 3.1$ during this training period to compute the Bayesian predictive distribution and estimate the probabilities for the largest expected earthquakes above magnitudes $m = 6.5$ and 7.3 to occur in the next $\Delta T = 5$, 10, and 15 days. This is illustrated in Supplementary Fig. 4.

**Sensitivity analysis of the Bayesian predictive distribution**. To check the sensitivity of the computed Bayesian predictive distribution with respect to the lower magnitude cutoff $m_c$, we fixed the forecasting time interval at $\Delta T = 10$ days and varied $m_c$. The results for the sequence with the target time interval $[T_s, T_e] =$

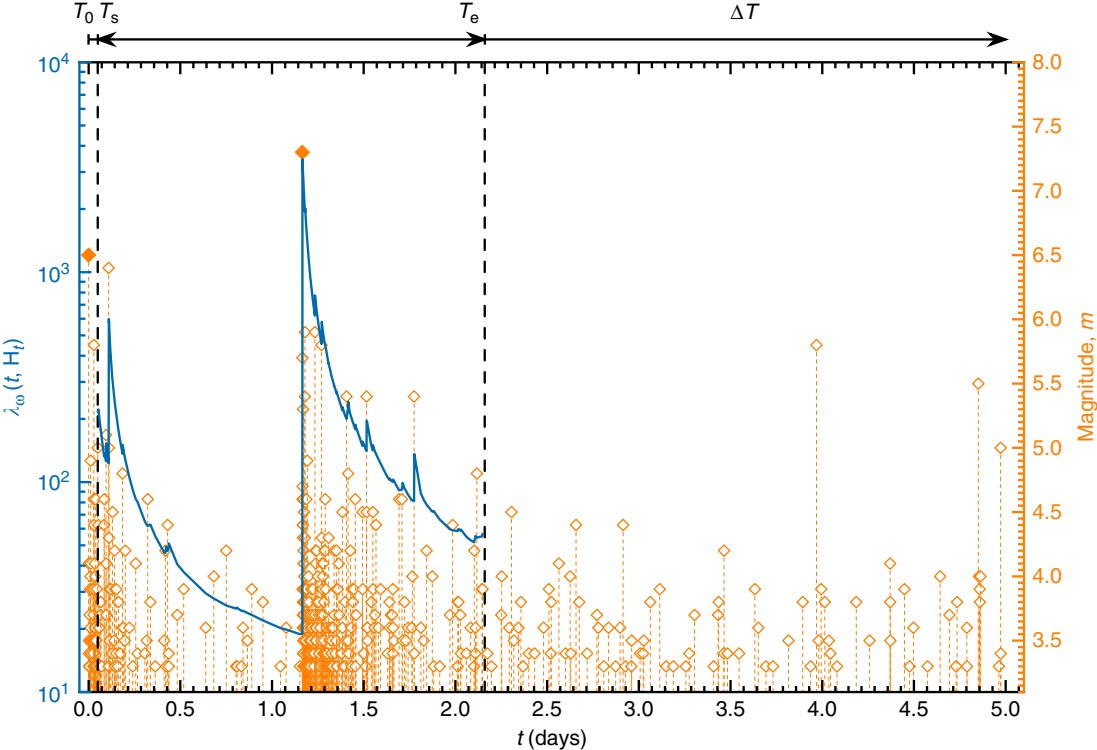

**Fig. 2** The 2016 Kumamoto earthquake sequence. The start of the sequence $T_0 = 0$ corresponds to the time of the occurrence of the M6.5 April 14, 2016, foreshock. All the events above magnitude $m_c \geq 3.3$ are shown. The Epidemic Type Aftershock Sequence (ETAS) model is fitted to the sequence during the training time interval $[T_0, T_e] = [0.0, 2.16]$ with $T_s = 0.05$ days. The estimated conditional earthquake rate, Eq. (7), (solid blue curve) is plotted using the following ETAS parameters: $\mu = 9.36$, $K = 0.67$, $c = 0.019$, $p = 1.27$, and $\alpha = 2.14$

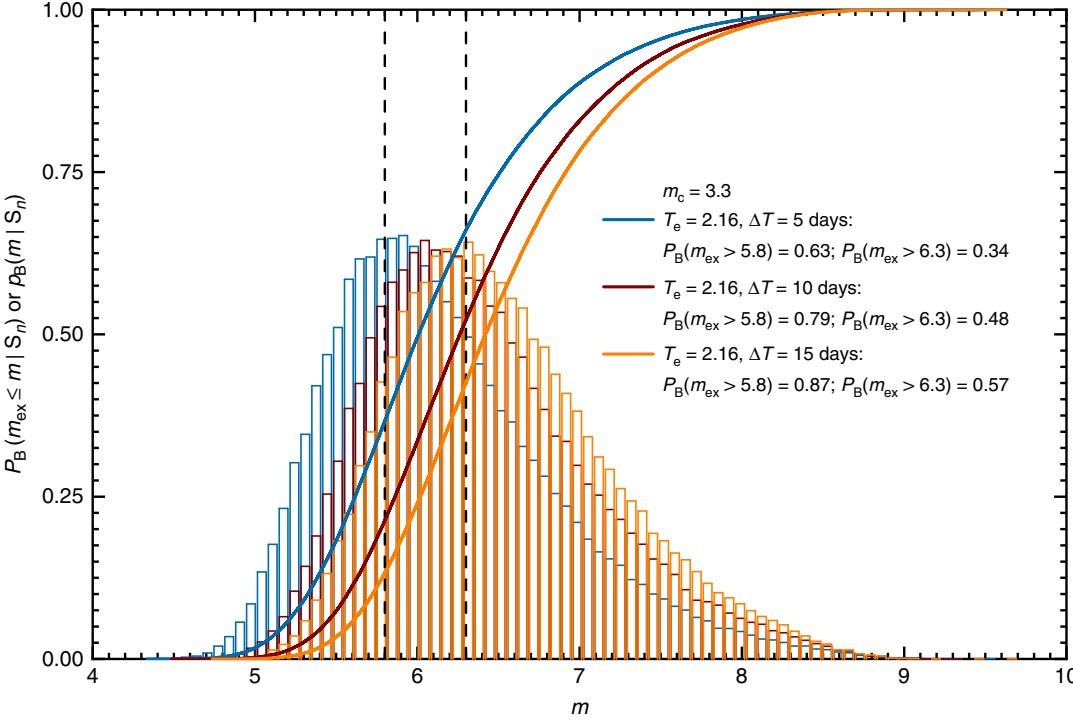

**Fig. 3** Bayesian predictive distribution. $P_B(m_{ex} \leq m|S_n)$, Eq. (1), are plotted as solid curves and the corresponding probability density functions $p_B(m|S_n)$, are plotted as histograms, for the sequence initiated by the April 14, 2016, M6.5 foreshock of the M7.3 Kumamoto earthquake (April 16, 2016). Each distirbution corresponds to the same early training time interval $[T_0, T_e] = [0.0, 2.16]$ with $T_s = 0.05$ days with all the events above magnitude $m_c \geq 3.3$ and for the different forecasting time intervals $\Delta T = 5$, 10, 15 days. The probabilities to have large earthquakes above magnitudes $m_{ex} \geq 5.8$ and $m_{ex} \geq 6.3$ are given in the legend

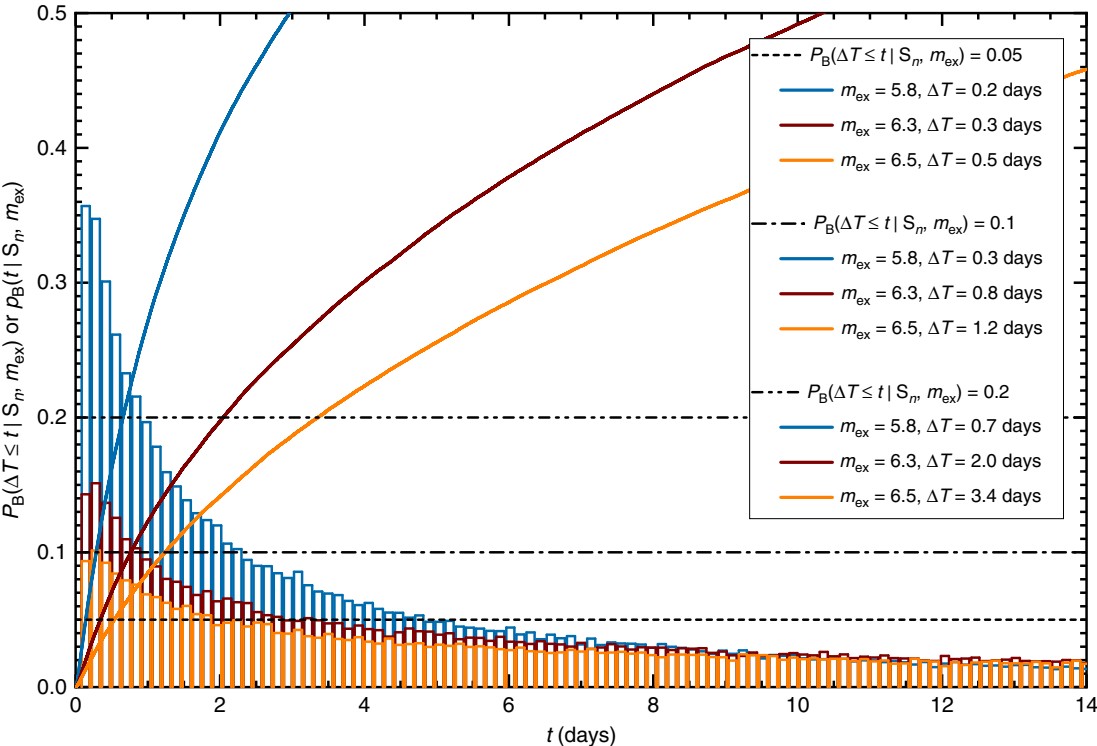

**Fig. 4** Bayesian predictive distribution. $P_B(\Delta T \leq t|S_n, m_{ex})$, Eq. (2), are plotted as solid curves and the corresponding probability density functions $p_B(t|M_n, m_{ex})$ are plotted as histograms, for the sequence initiated by the magnitude M6.5 Kumamoto foreshock (April 14, 2016). The results correspond to the same early training time interval from $T_s = 0.05$ to $T_e = 2.16$ days with all the events above magnitude $m_c \geq 3.3$ and for different forecasting extreme magnitudes $m_{ex} = 5.8$, 6.3, and 6.5. The horizontal lines correspond to the probability levels of 5% (dashed line), 10% (dash-dotted line), and 20% (dash-double-dotted line). The intercepts of these lines with the cumulative distribution functions give the forecasting or interarrival time intervals (provided in the figure legend) to the next largest expected earthquake with the magnitudes larger than $m_{ex}$

[0.05, 1.16] days and for $m_c = 3.1$, 3.3, and 3.5 are plotted in Supplementary Fig. 5a. Similarly, the sensitivity to the lower magnitude cutoff $m_c$ for the sequence with the target time interval $[T_s, T_e] = [0.05, 2.16]$ days is shown in Supplementary Fig. 5b for $m_c = 3.3$, 3.5, and 3.7. The results show relatively weak dependence on the lower magnitude cutoff.

In addition, we summarize the sensitivity of the obtained results with respect to the variability of the mean and variance of the prior distribution used for each model parameter. This is shown in Supplementary Fig. 6 for the sequence with the target time interval $[T_s, T_e] = [0.05, 2.16]$, the forecasting time interval $\Delta T = 10$ days, and the lower magnitude cutoff $m_c = 3.3$. The solid bold curve corresponds to the case of the noninformative flat prior. In Supplementary Fig. 6a we plot the results when the mean value of the Gamma prior corresponding to the parameter $\mu$ of the ETAS model is varied $\bar{\mu} = 0.01$, 0.1, 1.0, 4.0, 8.0, 10.0 and the rest of the mean values corresponding to the other parameters are fixed at $\{\bar{\beta}, \bar{K}, \bar{c}, \bar{p}, \bar{\alpha}\} = \{1.8, 0.7, 0.019, 1.27, 2.14\}$. In Supplementary Fig. 6b we plot the variability of the Bayesian predictive distribution due to changes in the mean value of the Gamma prior corresponding to the parameter $\bar{K} = 0.05$, 0.1, 0.5, 0.7, 1.0, 2.0.

Finally, we analyzed the effect of the functional form of the proposal distribution and the prior distribution on the obtained results. For the same sequence with the target time interval $[T_s, T_e] = [0.05, 2.16]$, the forecasting time interval $\Delta T = 10$ days, and the lower magnitude cutoff $m_c = 3.3$, we considered several combinations of the proposal and prior distributions. In addition to the truncated Normal distribution and the Gamma distribution, we used the lognormal distribution. The possible combinations of the proposal and prior distribution pairs are compared in Supplementary Fig. 7. It indicates that using either the truncated

Normal or the lognormal distributions as a proposal distribution produces statistically equivalent results. The choice of the prior distribution produces a small difference in the predictive distributions. This is an expected result in the Bayesian analysis as prior knowledge on model parameters can influence the posterior or predictive distributions.

**Comparison of the ETAS process with the Omori–Utsu law.** In Fig. 5, we compare the distributions for the magnitudes of the largest expected aftershock of the 2016 Kumamoto sequence to occur during the next $\Delta T = 10$ days using two training time intervals and several methods based on the approximation of the earthquake rate by either the ETAS process or the Omori–Utsu law. For the training time interval starting from the occurrence of the M6.5 foreshock and 1 day of aftershocks after M7.3 mainshock with $[T_0, T_e] = [0.0, 2.16]$ days, we plot the Bayesian predictive distributions computed using the ETAS process as the rate and the flat noninformative prior for the model parameters (solid blue curve) or the Gamma prior (solid dark brown curve). For the second time interval, we only considered the aftershocks during 1 day after the M7.3 mainshock. For this time interval, we plot the Bayesian predictive distribution with the ETAS rate with the Gamma prior for the model parameters (solid pink curve). In addition to using the ETAS model, we plot the Bayesian predictive distribution computed using the Omori–Utsu law for the earthquake rate as suggested in ref. [4] (dashed violet curve) and also using the MCMC sampling with the Gamma prior for the model parameters (dashed orange curve). Finally, we also plot the distribution to have a strong aftershock employing the extreme value (Gumbel) distribution by using the point estimates of the parameters $(\beta = 1.9, K = 54.0, c = 0.017, p = 1.04)$ of the

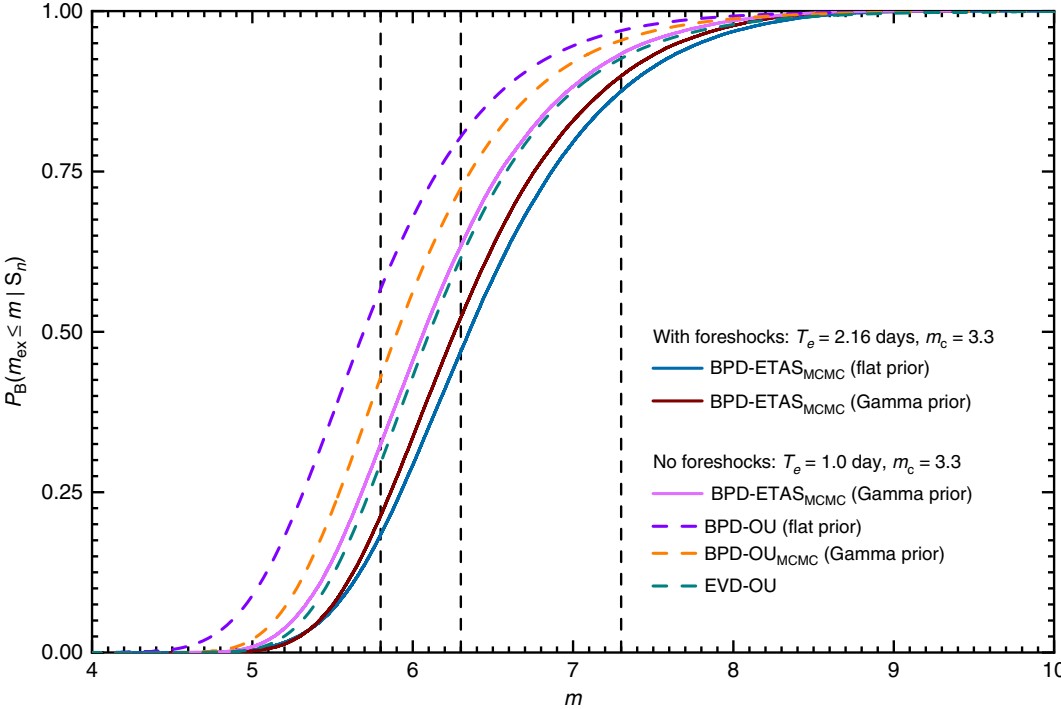

**Fig. 5** Comparison of the Bayesian predictive distributions. Several predictive distributions are plotted for the largest expected aftershock to be greater than $m$ during the next $\Delta T = 10$ days for the 2016 Kumamoto sequence. The distributions were computed employing: the ETAS rate and MCMC sampling with a flat prior (solid blue curve); the ETAS rate and MCMC sampling with the Gamma prior (solid dark brown curve) in both cases using the foreshocks and 1 day of aftershocks after the M7.3 mainshock. For the rest of the distributions, 1 day of aftershocks after the M7.3 mainshock was used. The distributions were computed using: the ETAS rate and MCMC sampling with the Gamma prior (solid pink curve); the Omori–Utsu (OU) rate with a flat prior (dashed violet curve); the OU rate and MCMC sampling with the Gamma prior (dashed orange curve); the Gumbel distribution and OU rate (dashed dark cyan curve)

**Table 1 Comparison of the probabilities to have the largest expected aftershock magnitude to be above $m_{ex}$ for the 2016 Kumamoto sequence computed using several methods**

| Probability | ETAS (flat) with foreshocks | ETAS (Gamma) with foreshocks | ETAS (Gamma) | OU | OU_MCMC | EVD |
|---|---|---|---|---|---|---|
| $P_B(m_{ex} > 5.8)$ | 0.82 | 0.79 | 0.68 | 0.45 | 0.59 | 0.71 |
| $P_B(m_{ex} > 6.3)$ | 0.53 | 0.48 | 0.37 | 0.21 | 0.30 | 0.38 |
| $P_B(m_{ex} > 7.3)$ | 0.13 | 0.10 | 0.07 | 0.03 | 0.05 | 0.07 |

In columns 2 and 3 the probabilities are computed using the earthquakes during the time interval $[T_0, T_e] = [0.0, 2.16]$, which includes the foreshocks and 1 day of aftershocks after the M7.3 mainshock: ETAS (flat) with foreshocks—the Bayesian predictive distribution using the ETAS model and a flat noninformative prior for the model parameters; ETAS (Gamma) with foreshocks—the Bayesian predictive distribution using the ETAS model and the Gamma prior. In columns 4–7 the probabilities are computed using only the aftershocks during 1 day after the M7.3 mainshock: ETAS (Gamma)—the Bayesian predictive distribution using the ETAS model and the Gamma prior; OU—the method developed in ref. [4] with the Omori–Utsu (OU) law and a flat prior; OU_MCMC—the Bayesian predictive distribution with the Gamma prior and using the OU law; EVD—the Gumbel distribution as the extreme value distribution with the parameters of the OU law estimated using the maximum-likelihood method

Omori–Utsu law (dashed dark cyan curve). This last plot is based on the Reasenberg and Jones approach[19]. The summary of the probabilities to have the largest expected aftershocks above certain magnitudes is given in Table 1. For both time intervals the earthquakes above magnitude $m_c \geq 3.3$ were used.

We point out to the difference between the two computed Bayesian predictive distributions using the ETAS process by considering two different training time intervals. The first predictive distribution (solid dark brown curve) was computed by incorporating the foreshocks and 1 day of aftershocks after the M7.3 mainshock. The second distribution (solid pink curve) only used 1 day of aftershocks. The inclusion of foreshocks resulted in higher probabilities for the occurrence of the largest expected aftershock during the forecasting time interval (see also Table 1). This result emphasizes the importance of the incorporation of the clustering nature of seismicity. The extreme value distribution

based on the Gumbel distribution with the Omori–Utsu rate (dashed dark cyan curve) is similar to the Bayesian predictive distribution based on the ETAS process with only 1 day of aftershock activity. This can be related to the fact that for this particular sequence the ETAS process and the Omori–Utsu model give similar estimates for the earthquake rate using only 1 day of aftershocks. The other two methods based on the Omori–Utsu law give lower probabilities of occurrence for the largest expected aftershock.

**Comparative analysis of several past prominent aftershock sequences**. In addition to the analysis of the Kumamoto sequence, we studied several other past aftershock sequences in order to compare the probabilities to have the largest expected aftershocks during the forecasting time interval. Specifically, we computed the

**Table 2 Comparison of the probabilities for the magnitudes of the largest expected aftershocks to be larger than $m = M_{ms} - 1.0$ for several prominent past aftershock sequences**

| Date | Name | $M_{ms}$ | $m_0$ | ETAS (Gamma) | OU | $OU_{MCMC}$ | EVD | $m_{as}$ |
|---|---|---|---|---|---|---|---|---|
| 2018/11/30 | Anchorage | 7.1 | 2.5 | 0.10 | 0.08 | 0.12 | 0.17 | 4.8 |
| 2016/11/13 | Kaikoura | 7.8 | 3.7 | 0.10 | 0.40 | 0.49 | 0.52 | 5.7 |
| 2016/04/16 | Kumamoto | 7.3 | 3.3 | 0.24 | 0.15 | 0.19 | 0.26 | 5.8 |
| 2011/03/11 | Tohoku | 9.0 | 5.3 | 0.27 | 0.15 | 0.19 | 0.26 | 6.7 |
| 2010/04/04 | El Mayor | 7.2 | 3.3 | 0.24 | 0.15 | 0.23 | 0.22 | 5.3 |
| 2008/06/14 | Iwate | 7.2 | 3.1 | 0.06 | 0.06 | 0.08 | 0.11 | 5.3 |
| 2007/03/25 | Noto | 6.9 | 3.1 | 0.15 | 0.13 | 0.17 | 0.23 | 4.9 |
| 2005/03/20 | Fukuoka | 7.0 | 3.1 | 0.02 | 0.01 | 0.02 | 0.03 | 5.4 |
| 2003/09/26 | Tokachi-oki | 8.0 | 3.3 | 0.27 | 0.23 | 0.28 | 0.40 | 6.5 |
| 2002/11/03 | Denali | 7.9 | 3.0 | 0.08 | 0.24 | 0.38 | 0.47 | 5.5 |

The Bayesian predictive distribution $P_B(m_{ex} > m|S, \Delta T)$ was computed using: ETAS—the ETAS model with the Gamma prior for the model parameters; OU—the method developed in ref. [4] with the Omori–Utsu law and a flat prior; $OU_{MCMC}$—the MCMC sampling with the Omori–Utsu law and the Gamma prior. These are compared with the probabilities computed using the Gumbel distribution as the EVD for the non-homogeneous Poisson process with the parameters of the Omori–Utsu law estimated using the maximum-likelihood method. The training time interval $[T_0, T_e] = [0.0, 2.0]$ days and the forecasting time interval $\Delta T = 10$ days were used, where $T_0 = 0$ corresponds to the time of the occurrence of the mainshocks. The last column gives the magnitudes of the actual largest aftershock that occurred during the forecasting time interval $\Delta T = 10$ days

Bayesian predictive distributions for each sequence using the ETAS approximation and assuming a Gamma prior for the model parameters. In addition, we also considered the Omori–Utsu approximation for the earthquake rate and assumed a flat prior distribution[4]. We also considered the approach suggested in this work but we replaced the ETAS model with the earthquake rate approximated by the Omori–Utsu law. Finally, we considered the model, which is equivalent to the Reasenberg and Jones approach[19], with the parameters of the Omori–Utsu law estimated for each individual sequence. The last model assumed that the probabilities were computed using Eq. (11) and the estimated values of the Omori–Utsu parameters. For all these computations, we considered the training time interval $[T_0, T_e] = [0.0, 2.0]$ days and the forecasting time interval $\Delta T = 10$ days, where $T_0 = 0$ corresponds to the time of the occurrence of each mainshock. The summary of the results is given in Table 2.

## Discussion

The obtained results indicate that the proposed method computes robustly the Bayesian predictive distribution for the magnitude of the largest expected earthquake to be above a certain magnitude and during a specified forecasting time interval. It also allows the estimation of the interarrival time interval during which the largest expected earthquake above a certain magnitude can occur at a given probability level.

The MCMC method was used to sample the posterior distribution, Eq. (3). The sampling generates the Markov chains of the model parameters that allows the estimation of their variability. The MCMC chain for the ETAS parameters generated for the studied sequence of events is given in Supplementary Figs. 1–3. Particularly, in Supplementary Fig. 2, we provide the distribution of the model parameters for this sequence and report the 95% Bayesian credibility intervals for each model parameter. This differs from a standard practice, where the model parameters are estimated using the maximum-likelihood approach, and are used, for example, in the computation of the extreme value distribution.

The ETAS model generates a stochastic sequence of events, where magnitudes are drawn from the exponential distribution, Eq. (6). However, when comparing the Bayesian predictive distribution constructed from the ensemble of realizations of the ETAS process using the MCMC chain of the model parameters, with the Bayesian predictive distribution using the Gumbel distribution, Eq. (11), one observes a statistically significant deviation between these two distributions (Supplementary Fig. 8). The

Kolmogorov–Smirnov test showed that the null hypothesis that these two distributions came from the same distribution was rejected at the 1% significance level. This is due to the fact that the probability that the magnitude $m_{ex}$ of the maximum event in the forecasting time interval $[T_e, T_e + \Delta T]$ is greater than $m$ cannot be expressed by a closed formula, because the conditional intensity function Eq. (7) changes stochastically with each new event during the forecasting time interval. This differs from a stationary or non-stationary Poisson processes, where the event rate $\lambda(t)$ is a deterministic function of time, and thus the calculation of the extreme value distribution using the theorem of total probability becomes straightforward. However, in Supplementary Note 1, we show that the analytic calculation of the extreme value distribution for the ETAS process (or a Hawkes process in general) leads to an integral equation that, in principle, can be evaluated numerically.

The prior distribution for the model parameters plays an important role in the Bayesian analysis. In our implementation, we used the Gamma distribution. However, we also considered the lognormal distribution. We examined the influence of the initial knowledge of the model parameters on the final predictive distributions. The distributions of the model parameters $\{\beta, K, c, p, \alpha\}$ in the MCMC chain are reasonably stable with respect to the changes of the mean and variance of the prior distribution. On the other hand, the background rate $\mu$ is strongly affected by the choice of the mean and variance of the prior distribution for this specific sequence we analyzed (Supplementary Fig. 6). This can be explained by the relative flatness of the likelihood function associated with the $\mu$ parameter. As a result, the convergence of the Markov Chain is strongly affected by the prior distribution.

The formulated approach was applied, in detail, to the 2016 Kumamoto, Japan, earthquake sequence. This sequence had a complex structure with well-observed foreshocks, the mainshock and subsequence aftershocks. We were able to compute successfully the probabilities of having strong earthquakes to occur during the future time intervals by analyzing two sequences of events of varying length. The first sequence comprised the first day of aftershocks of the mainshock in addition to the 1.16 days of foreshocks. In this case, we estimated the probabilities of having large aftershocks above magnitudes 5.8 and 6.3 to occur during the fixed future time intervals (Fig. 3). The second sequence comprised only the foreshocks of the M7.3 mainshock and we estimated the probability of having a mainshock above magnitude 7.3 to occur in the next $\Delta T = 5$, 10, and 15 days (Supplementary Fig. 4). The obtained probability for the occurrence of the M7.3 mainshock is rather small for such large

earthquakes. We think that this is limited by the information content the seismicity rate provides. To improve such forecasting, one needs to consider, in addition, other physics-based processes, for example, the stress transfer, geometry and distribution of faults, and more accurate rheological description of the crust and mantle.

The method differs from several past studies[4,19,20] by using the ETAS model instead of the Omori–Utsu law for the conditional earthquake rate. To illustrate this, we compared the method suggested in this work to several approaches of computing the probabilities for the magnitudes of the largest expected aftershocks based on the Omori–Utsu approximation of the earthquake rate. Specifically, we used the Gumbel distribution for the extreme value distribution and the point estimates for the Omori–Utsu parameters, which is the model proposed by Reasenberg and Jones[19]. The comparison of the obtained results for several past prominent aftershock sequences (Table 2), suggests that the method based on the Gumbel distribution and the point estimates of the model parameters provides typically higher probabilities for the occurrence of the largest expected aftershocks than the methods based on the Bayesian predictive statistics. For this comparison, we used only aftershocks during 2 days after each mainshock. However, the inclusion of the complicated clustering structure of seismicity can result in higher probabilities for the occurrence of the largest expected aftershocks, as illustrated by the 2016 Kumamoto sequence (Table 1).

One of the important advantages of the implemented method is that it fully incorporates the uncertainties of the model parameters into the analysis and the clustering structure of seismicity. To account for these uncertainties, we used the MCMC method to sample the posterior distribution of the model parameters (Eq. (3)). We found that the ETAS-based method deviates clearly from other approaches, e.g. non-homogeneous Poisson process driven by the Omori–Utsu rate (Fig. 5). This demonstrates that complex triggering including foreshocks and/or higher-order aftershocks cannot be neglected for purposes of earthquake/aftershock forecasting.

## Methods

**Computation of the Bayesian predictive distribution**. The direct computation of the Bayesian predictive distribution, Eq. (1) or (2), is prohibitive for several reasons. First, the integration is performed over a multidimensional parameter space $\{\Theta, \Omega\}$. In case of the parametric model, Eqs. (6) and (7), it has total of six parameters. Second, the evaluation of the ETAS productivity, $\Lambda_\omega(\Delta T)$, during the forecasting time interval $[T_e, T_e + \Delta T]$ has to be performed by simulating the ETAS model forward in time using the ETAS parameters sampled from the posterior distribution (Eq. (3)). Third, the functional form of the extreme value distributions $P_{EV}(m_{ex} > m|\theta, \omega, \Delta T)$ or $P_{EV}(\Delta T \le t|\theta, \omega, m_{ex})$ is typically known for i.i.d. random variables.

For the ETAS model, the distribution of the magnitudes of extreme events deviates from the Gumbel distribution. For the above reasons, the Bayesian predictive distribution can be estimated by employing the MCMC sampling procedure to generate Markov chains of the ETAS model parameters $\{\Theta^{(i)}, \Omega^{(i)}\}$, $i = 1, …, N_{sim}$ drawn from the posterior distribution[3,47]. To sample the posterior distribution, Eq. (3), we use the Metropolis-within-Gibbs algorithm. To compute the Bayesian predictive distribution one can generate sequences of the stochastic point process under consideration and extract the maximum magnitude event from each sequence. This will allow one to construct the empirical distribution function for the largest magnitudes that approximates the Bayesian predictive distribution. In addition, this allows to incorporate the uncertainties of the model parameters into the estimation of the predictive distributions.

**MCMC sampling**. To sample the posterior distribution, Eq. (3), we used the Metropolis-within-Gibbs algorithm[47,48]. In this algorithm, the updates of the parameters are performed one at a time during each Markov step. The proposal distribution is univariate and the samples are drawn from the conditional distribution. The Metropolis–Hastings ratios are defined as follows:

$$\mathrm{MH}_{\theta, \omega} = \frac{p(\tilde{\theta}, \tilde{\omega}|M_n)}{p(\theta, \omega|M_n)} \frac{J(\theta, \omega|\tilde{\theta}, \tilde{\omega})}{J(\tilde{\theta}, \tilde{\omega}|\theta, \omega)}$$

$$= \frac{L(M_n|\tilde{\theta}, \tilde{\omega})\pi(\tilde{\theta}, \tilde{\omega})}{L(M_n|\theta, \omega)\pi(\theta, \omega)} \frac{J(\theta, \omega|\tilde{\theta}, \tilde{\omega})}{J(\tilde{\theta}, \tilde{\omega}|\theta, \omega)}, \quad (8)$$

where $\tilde{\theta}, \tilde{\omega}$ indicate the proposed values of the model parameters. This ratios are applied individually for each model parameter when generating the corresponding Markov chains. The truncated Normal distribution is used for the proposal distribution, $J(\theta, \omega|\tilde{\theta}, \tilde{\omega})$, to constrain the parameters to the positive values. For the prior distribution, $\pi(\theta, \omega)$, the Gamma distribution is used.

The Metropolis-within-Gibbs sampler combines two MCMC methods: the Gibbs sampler and the Metropolis–Hastings algorithm[3,47]. The Gibbs sampler can be used to generate random variables from multivariate distributions by drawing each random variable component from a univariate fully conditional distribution one at a time. To use the Gibbs sampler alone, one needs to generate efficiently the random samples from the fully conditional univariate distributions constructed from the original multivariate distribution under consideration. When this is not possible one can substitute this step by drawing the univariate samples using the Metropolis–Hastings algorithm. In Algorithm 1, we outline the Metropolis-within-Gibbs sampler, which was used in our analysis. The Metropolis–Hastings algorithm is given in Algorithm 2.

Algorithm 1
The Metropolis-within-Gibbs (MwG) algorithm
1: **procedure** METROPOLIS-WITHIN-GIBBS ($\mathbf{w}_0$, $N_{sim}$)
2:   $\mathbf{m}^{(0)} \leftarrow \mathbf{w}_0$                ▷ set a starting value for the chain
3:   **for** $k = 1 : N_{sim}$ **do**            ▷ iterate $N_{sim}$ times
4:     $\mathbf{p} \leftarrow \mathbf{m}^{(k-1)}$            ▷ the vector of parameters at $(k-1)$th step
5:     **for** $n = 1 : N_{par}$ **do**          ▷ iterate $N_{par}$ times
6:       $\mathbf{m}^{(k)}(n) \leftarrow$ METROPOLIS-HASTINGS($\mathbf{m}^{(k-1)}(n)$, $\mathbf{p}$, 1)
7:       $\mathbf{p}(n) \leftarrow \mathbf{m}^{(k)}(n)$       ▷ update the value to the newly generated sample
8: **return m**                       ▷ return the chains for all parameters

The Metropolis-within-Gibbs sampler (Algorithm 1) takes as input the initial values of the model parameters $\mathbf{w}_0 = \{\beta, \mu, K, c, p, \alpha\}$ and the number of samples to generate $N_{sim}$. During each iteration of the main loop (Line 3) it updates each parameter of the model individually by calling the Metropolis–Hastings sampler (Algorithm 2). This is done in the second loop (Lines 5–7). The parameters are updated one at a time from the list $\mathbf{p} = \{\beta, \mu, K, c, p, \alpha\}$. When updating the individual parameters (Line 7) the vector $\mathbf{p}$ is updated by a newly generated sample of each parameter. The Metropolis-within-Gibbs algorithm allows to sample multidimensional posterior distributions by sampling each parameter individually. During the sampling it generates the Markov chain of the parameters by exploring the parameter space given by the posterior distribution.

The Metropolis–Hastings sampler takes as input the initial value $v_0$ of a given model parameter, the full vector of model parameters $\mathbf{p}$, and the number of iterations $N = 1$. The Metropolis–Hastings sampler performs only a single sampling ($N = 1$) of a given parameter while the other parameters are kept fixed. So effectively it performs MCMC sampling of a univariate distribution. It generates a new candidate from a proposal distribution $J(y, x)$ (Line 4). Then it computes the acceptance ratio $r = \mathrm{MH}_\mathbf{p}$ using Eq. (8) (Line 5). If $\min(1, r) \ge u$, where $u$ is a random number generated from a uniform distribution $\mathcal{U}(0, 1)$, it accepts a new candidate, otherwise it does not update the parameter. The Metropolis–Hastings ratios, $\mathrm{MH}_\mathbf{p} = \mathrm{MH}_{\theta, \omega}$, are given in Eq. (8).

Algorithm 2
The Metropolis–Hastings (MH) algorithm
1:**procedure** METROPOLIS–HASTINGS ($v_0$, $\mathbf{p}$, $N$)
2: $\mathbf{x}^{(0)} \leftarrow v_0$                ▷ set a starting value for the chain
3: **for** $i = 1 : N$ **do**              ▷ iterate $N$ times
4:   $\tilde{\mathbf{x}} \leftarrow J(y|\mathbf{x}^{(i-1)})$       ▷ generate a candidate value from the proposal distribution
5:   $r \leftarrow \mathrm{MH}_\mathbf{p}$            ▷ compute the acceptance ratio
6:   $u \leftarrow \mathcal{U}(0, 1)$           ▷ generate a uniform random number
7:   **if** $\min(1, r) \ge u$ **then**
8:     $\mathbf{x}^{(i)} \leftarrow \tilde{\mathbf{x}}$          ▷ accept the candidate $\tilde{\mathbf{x}}$
9:   **else**
10:    $\mathbf{x}^{(i)} \leftarrow \mathbf{x}^{(i-1)}$         ▷ reject the candidate $\tilde{\mathbf{x}}$
11: **return x**                   ▷ return a single chain

The MCMC chains of the ETAS model parameters were generated for total of $N_{sim} = 200,000$ steps and 100,000 steps were discarded as burn-in. Particular generated chains for each model parameter are shown in Supplementary Fig. 1 using the sequence of the 2016 Kumamoto events using the training time interval $[T_s, T_e] = [0.05, 2.16]$ and the lower magnitude cutoff $m_c = 3.3$.

The Gamma distribution was used as a prior for the model parameters, $\pi(\{\theta, \omega\}) = g(\beta)g(\mu)g(K)g(c)g(p)g(\alpha)$:

$$g(x|a, b) = \frac{1}{b^a \Gamma(a)} x^{a-1} \mathrm{e}^{-x/b}, \quad (9)$$

where $\Gamma()$ is a Gamma function, $a$ is a shape parameter and $b$ is a scale parameter. The parameters of the Gamma distribution, $a$ and $b$, can be expressed in terms of the mean and the variance of the Gamma distribution:

$$a = \frac{\mathbb{E}(x)^2}{\mathrm{Var}(x)}, \qquad b = \frac{\mathrm{Var}(x)}{\mathbb{E}(x)}. \quad (10)$$

The truncated Normal distribution was used as a proposal distribution, $J(y|x)$, where parameters $y$ and $x$ stand for the mean and standard deviation of the truncated Normal distribution, respectively. The parameters of the prior distributions and the variance of the proposal distribution are provided in Table 3

**Table 3 Summary of the parameters used for the prior distribution and the proposal distribution corresponding to the parameters of the ETAS model $\{\theta, \omega\} = \{\beta, \mu, K, c, p, \alpha\}$**

| Parameter | $\beta$ | $\mu$ | $K$ | $c$ | $p$ | $\alpha$ |
|---|---|---|---|---|---|---|
| The sequence: $[T_s, T_e] = [0.03, 1.16]$ | | | | | | |
| Mean (Gamma) | 1.7 | 1.0 | 0.7 | 0.019 | 1.27 | 2.1 |
| Var (Gamma) | 0.1 | 0.05 | 0.01 | 0.0005 | 0.05 | 0.1 |
| Var (TN) | 0.05 | 0.05 | 0.05 | 0.001 | 0.01 | 0.01 |
| The sequence: $[T_s, T_e] = [0.05, 2.16]$ | | | | | | |
| Mean (Gamma) | 1.8 | 9.4 | 0.67 | 0.019 | 1.27 | 2.14 |
| Var (Gamma) | 0.1 | 0.1 | 0.01 | 0.0001 | 0.05 | 0.1 |
| Var (TN) | 0.05 | 0.05 | 0.05 | 0.001 | 0.01 | 0.01 |

For the priors $\pi(\{\theta, \omega\})$ the Gamma distribution was used with the mean and variance specified for each parameter. For the proposal distribution $J(y|x)$ the truncated normal (TN) distribution was used with the specified variance for each model parameter

The distributions of each ETAS parameter from the MCMC chains (Supplementary Fig. 1) are shown in Supplementary Fig. 2. The solid curves represent the prior distribution for each model parameter. Supplementary Fig. 3 shows the cross plots for each pair of the parameters. Some of the parameters show strong correlation.

**The extreme value distributions for the ETAS process**. For sequences of earthquakes, for which event magnitudes are i.i.d. random variables, the probability that the maximum event magnitude is greater than $m$ for all possible number of events during a given time interval $[T_e, T_e + \Delta T]$ is[1,2,42,44,45]

$$P_{EV}(m_{ex} > m|\theta, \omega, \Delta T) = 1 - \exp\{-\Lambda_\omega(\Delta T)[1 - F_\theta(m)]\}, \quad (11)$$

where the productivity is $\Lambda_\omega(\Delta T) = \int_{T_e}^{T_e + \Delta T} \lambda_\omega(t|\mathcal{H}_t) dt$. Using the exponential model for the magnitude distribution, which is considered in this work, this results in the Gumbel distribution. Equation (11) was used by Reasenberg and Jones[19] to formulate the model for the distribution of extreme aftershocks in California assuming that the aftershock rate can be approximated by the Omori–Utsu law.

If the functional form of the extreme value distribution is known, the Bayesian predictive distribution, Eq. (1), at a given magnitude $m$ can be computed by using the MCMC chain of the parameters of the ETAS model. Each sample of the chain $\{\Theta^{(i)}, \Omega^{(i)}\}$ can be used to estimate the productivity of the ETAS model $\Lambda_\omega(\Delta T)$ during the future forecasting time interval $[T_e, T_e + \Delta T]$ by performing the stochastic simulation of the ETAS process. Using the obtained productivity, one can compute the value of the extreme value distribution for a given magnitude $m$.

The extreme value distribution can be also computed empirically by generating the ensemble of earthquake sequences by using the ETAS process and extracting the maximum magnitude from each sequence and constructing the empirical distribution function for the extreme magnitudes. We performed such calculations when estimating the Bayesian predictive distribution in this work. The ETAS model was simulated using the thinning algorithm[42]. In Supplementary Fig. 8, we compare this empirically computed predictive distribution with the one computed using the extreme value distribution given in Eq. (11). For this latter case, we performed ensemble simulation of the ETAS processes with the parameters taken from the MCMC chain, which were sampled from the posterior distribution. For each parameter sample, the productivity of the ETAS model $\Lambda_\omega(\Delta T)$ was approximated by the number of events above magnitude $m_c$ generated in the stochastic simulation of the ETAS model. The two distributions given in Supplementary Fig. 8 exhibit a statistically significant deviation. Therefore, the extreme value distribution given in Eq. (11) is not applicable to the ETAS process. We provide the analytical derivation of the extreme value distribution for the ETAS process in Supplementary Note 1.

## Data availability

The earthquake catalogs were downloaded from: the Japan Meteorological Agency (http://www.jma.go.jp/en/quake/); Hauksson et al.[49] waveform relocated catalog for Southern California (http://web.gps.caltech.edu/~hauksson/catalogs/index.html); Alaska Earthquake Information Center (http://www.aeic.alaska.edu/htmldocs/db2catalog.html); New Zealand GEONET (http://quakesearch.geonet.org.nz/).

The computed data used to plot Figs. 2–5, Supplementary Figs. 1 and 4 are provided in Source Data.xlsx file.

## Code availability

The computer code to perform the analysis can be requested from the authors.

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

## Acknowledgements

R.S. was supported by the NSERC Discovery grant. J.Z. was partially supported by Grants-in-Aid No. 19H04073 for Scientific Research from the Japan Society for the Promotion of Science. G.Z. was supported by the DFG Collaborative Research Centre 1294 (Data Assimilation—The seamless integration of data and models, project B04).

## Author contributions

R.S., J.Z., G.Z., Y.O. were involved in the formulation of the problem and the design of the methodology. R.S. wrote the computer code to analyze earthquake data and performed the statistical analysis. All the authors participated in the discussions and contributed to the writing of the manuscript.

## Additional information

**Competing interests:** The authors declare no competing interests.

