## [Peer Review File · Nature Communications]

Reviewers' comments:

Reviewer #1 (Remarks to the Author):

The paper performs Bayesian inference for the maximum magnitude in a given forecasting interval, where the occurrence of events is described by an Epidemic Type Aftershock Sequence (ETAS) point process. The ETAS models are often used for forecasting the spatio-temporal clustering of aftershocks in the literature. However, estimating the parameters of an ETAS model for an ongoing sequence, remains a challenge given the evolutionary functional form of the ETAS model (every event with magnitude larger than the cut-off level contributes to the seismicity rate), its number of parameters (6 parameters in this paper) and the initial incompleteness of the catalogue. The paper is well-written and well-organized. The reviewer recommends publication after the authors have revised the manuscript to address specific concerns.

(1) The novelty of the paper is in sampling the ETAS parameters using MCMC: Gibbs/Metropolis sampler --conditioned on the events that have already taken place in the sequence-- in order to derive a posterior distribution for the maximum magnitude that is expected to happen in a given forecasting interval. To reviewer's knowledge, the use of MCMC/Metropolis-Hastings (but not Gibbs Sampler) for sampling the ETAS parameters given the events that have already taken place in the sequence is investigated in another paper (co-authored by the reviewer and cited in the manuscript):

Ebrahimian, H., & Jalayer, F. (2017). Robust seismicity forecasting based on Bayesian parameter estimation for epidemiological spatio-temporal aftershock clustering models. *Scientific reports*, 7(1), 9803.

In Ebrahimian and Jalayer (2017), the probability of having events with magnitude larger than certain thresholds in a given forecasting interval is reported using a formulation similar to Eq. S1 of the paper. However, Ebrahimian and Jalayer (2017) did not derive the distribution of maximum magnitude analytically (Section S1.1) and non-parametrically as is done in this manuscript. The manuscript seems to do a thorough analysis of the literature. However, it would be useful to indicate more explicitly the previous work done by Ebrahimian and Jalayer (2017) and highlight the differences.

(2) The paper is of interest to others in the field. Quasi real-time tuning of ETAS models has been a challenge especially in its more general spatio-temporal clustering format. Nevertheless, since the objective of the paper is estimating the maximum magnitude, the time-magnitude version is adopted in the manuscript. The MCMC framework provides the possibility of sampling directly from a conditional distribution (the joint probability distribution of the ETAS model parameters given the events that have already taken place in the sequence) which is known but for a constant. This possibility can be also useful for Bayesian inference of other quantities such as the rate of seismicity.

(3) Another point of interest of crucial importance, beyond the technical difficulties associated to parameter estimation, is whether/under which circumstances the ETAS model is able to forecast with a certain confidence events that are larger in magnitude with respect to the main event or to forecast a mainshock based on its foreshocks (if they exist). With regard to this point, the results of forecasting after $t=1.16$ days (shown in the supplementary material Fig. S5) apparently indicate that the forecasts provided for $M>7.3$ right before it took place did not foresee it with a large probability (obviously the interpretation is relative). It would be important to provide some insight with this regard in the discussion of the results.

(4) It would be useful to shed some more insight to the assumptions made for deriving the likelihood in Eq. (4) and how it is implemented. Given that the Bayesian inference based on MCMC is used and given that the prior distributions are assumed to be independent, it is the likelihood

function that determines to a large degree the possible correlations between the inter-arrival times and the magnitudes. The formulation in Eq. 4 apparently seems to indicate that the inter-arrival times are expressed as corresponding to a Poisson point process (and therefore independent) and the magnitudes are i.i.d. and also independent from the inter-arrival times. Moreover, the formulation seems also to be based on the assumption that the conditional rate has a functional form that is not going to change during the sequence (non-evolutionary). This somehow seems to be confirmed by the very small difference observed (Fig. S7) between the distribution of maximum magnitude based on Eq. S1 and the non-parametric one based on the samples generated from MCMC-Gibbs sampling.

(5) When sampling the ETAS parameters, does the manuscript consider the triggering effect of the events that are going to take place in the forecasting interval? Or the paper assumes that the ETAS rate considers only the triggering of the events taken place before the forecasting interval? This latter assumption is going to simplify the simulation process and would probably affect the results in a non-conservative manner. Please specify in the paper.

(6) The analytical formulation shown in Section S1.1 is very interesting. It would be curious to see how the results are going to differ from those calculated based on Equation S1. Nevertheless, the paper does not seem to use or demonstrate the implementation/results of the formulations derived in S1.1 --which seem to have a recursive formulation and might be time consuming to calculate. There is a reference to this section in Page 10, line 227 but it was not very clear to the reviewer. Even if the formulation is challenging to be implemented, it would be useful and insightful to mention those challenges.

Minor Comments:

(1) Page 7, line 148: The manuscript adopts the truncated Normal for sampling positive values from the proposal distribution, why Lognormal is not used?

(2) The two algorithms reported in the Methods sections were not very easy to follow for the reviewer. Please either explain in more detail the notation used or replace them with less detailed and more conceptual diagrams.

(3) Page 3, Line 74: What is the upper bound for i ?

(4) Is Equation S2 providing the numerical Monte Carlo estimate of Eq 1? If this is the case, please specify. The paragraph following Eq. S2 was not very clear with this regard.

(5) Page 3 supplementary material: by complete probability it is intended "total probability"?

(6) Figure S9c what are the transformed time and earthquake index? Please explain.

05/02/2019

Fatemeh Jalayer

University of Naples, Federico II

Reviewer #2 (Remarks to the Author):

This review was solicited to assess the impact and importance of this manuscript, and is thus not a full technical review.

I think this manuscript describes an important advance towards estimating the maximum expected magnitudes within an operational earthquake forecast. The key advance is that rather than simply extrapolating a Gutenberg-Richter distribution from an Omori-law rate, a more comprehensive set of ETAS rates are used, which captures interactions and secondary triggering.

The authors have developed an innovative approach to solving the Bayesian equations by sampling with Markov chains.

In my opinion, the authors could improve the manuscript primarily with more descriptive language, and by using more direct comparisons between this method and past practice (I did find this in the supplements, but I think it should be in the main body of the paper).

Further examples include:

(1) "...an earthquake sequence governed by the ETAS process" I think it is important for a general audience to fully describe ETAS, and clarify that it is a statistical method that does not govern, but attempts to describe physics through empiricism.

(2) Early in the draft it is stated: "In addition, we show that the extreme value distribution for the magnitudes of the largest events in the ETAS model deviates from the Gumbel distribution indicating a certain degree of correlation between event magnitudes"

This is an important result that shows why your work is important, and people may not grasp this without a definition of the Gumbel distribution. I'd suggest a couple more lines here using plain language.

(3) "For the prior distribution, $p(q;w)$, of the model parameters, we used a Gamma distribution (specific values for the mean and variance of the priors for each model parameter are provided in Table S1). For the proposal distribution $J(x|x')$, we used a truncated normal distribution ..."

I'm sure these are the right choices, but I think it detracts a little from the paper when certain distributions/parameters are used without stating why, nor what the impact would be to use different choices. That said, for the most part the sensitivity analyses are commendable. I'm just suggesting some additional text in these cases.

(4) Results: "For a fixed probability level of 5% the estimated interarrival times are $DT = 0:26, 0:81, \text{ and } 1:38$ days, respectively. For a 10% probability level the estimated interarrival times are $DT = 0:56, 2:26, \text{ and } 4:52$ days."

It's probably worthwhile to place results like these into forecast context, and in so doing maybe drop the significant figures back a little.

Last point: I wonder how this method would work in a region where there are very large faults. Would the exponential magnitude-frequency distribution require modification to account for background probability of very high magnitudes on these structures that might originate from outside of the region where the ETAS parameters apply?

Reviewer #3 (Remarks to the Author):

The manuscript "Bayesian Inference on the Magnitude of the Largest Expected Earthquake" by Shcherbakov and colleagues aims at estimating the probability of extreme earthquakes to be above a certain magnitude. This is done in aftershock time series by applying Bayes theorem to the epidemic-type aftershock sequence (ETAS) model. The work is sound and the paper well

written.

However, I fail to see much novelty. It mostly represents some incremental research on topics and methods already well established:

- 1) Dozens of articles if not hundreds use ETAS as baseline model.
- 2) Bayesian approaches to estimate M_{max} are already available in the literature, as already stated by the authors.

Although I'm not certain that Bayes' Theorem has not already been applied for ETAS, the new implementation will mostly interest statisticians. How this new approach performs compared to non-epidemic type models has yet to be proven and this will require model comparisons for numerous earthquake sequences. Here the study case is only illustrative in nature. Moreover, in operational earthquake forecasting (OEF), first responders can already use simple Omori statistics (or variants such as STEP) to estimate large aftershock probabilities. They can then reiterate to model subsequent aftershocks. In view of the large number of parameters required for ETAS, I'm not so sure that the added uncertainties will help improve OEF.

Finally, when discussing about "extreme earthquakes", the authors should be more cautious. ETAS, in this context, is misleading since it represents a point process while large earthquakes occur on existing faults, whose geometry and loading state are not included in ETAS. Therefore, the model is unlikely to be representative of the ongoing physical processes in the crust.

To conclude, I believe that this paper is not original enough nor of sufficient importance to be published in a high-impact journal. It is also technical in nature and will not interest a wide readership. Better journals for such a work would be JGR, BSSA, or similar.

Reply to the reviewer comments:

We would like to thank the reviewers for their constructive comments and suggestions. Those comments helped to improve and clarify the presentation and results reported in the revised version of the paper.

Reply to the review of the first referee:

We would like to thank Dr. Jalayer for her thoughtful and constructive comments. These comments raised important points that we would like to clarify. We also would like to thank the reviewer for her suggestions to improve the text and overall presentation of the paper.

“(1) The novelty of the paper is in sampling the ETAS parameters using MCMC: Gibbs/Metropolis sampler --conditioned on the events that have already taken place in the sequence-- in order to derive a posterior distribution for the maximum magnitude that is expected to happen in a given forecasting interval. To reviewer’s knowledge, the use of MCMC/Metropolis-Hastings (but not Gibbs Sampler) for sampling the ETAS parameters given the events that have already taken place in the sequence is investigated in another paper (co-authored by the reviewer and cited in the manuscript): Ebrahimian, H., & Jalayer, F. (2017). Robust seismicity forecasting based on Bayesian parameter estimation for epidemiological spatio-temporal aftershock clustering models. Scientific reports, 7(1), 9803. In Ebrahimian and Jalayer (2017), the probability of having events with magnitude larger than certain thresholds in a given forecasting interval is reported using a formulation similar to Eq. S1 of the paper. However, Ebrahimian and Jalayer (2017) did not derive the distribution of maximum magnitude analytically (Section S1.1) and non-parametrically as is done in this manuscript. The manuscript seems to do a thorough analysis of the literature. However, it would be useful to indicate more explicitly the previous work done by Ebrahimian and Jalayer (2017) and highlight the differences.”

Reply:

In the revised version of the paper, we added a paragraph to Introduction, where we provided more details about the above-mentioned paper [EJ2017]. In that work, the spatio-temporal version of the ETAS model is used. However, several constraints on the model parameters were introduced, specifically that, $\alpha = \beta$, $p > 1$, and the parameters K is constrained on the number of possible earthquakes. As a result, the total number of the ETAS parameters sampled is 5. Whereas, in our approach we consider the unconstrained ETAS parameters. Although, we acknowledge that we use only time dependent version of the model. Originally, we implemented the Metropolis-Hastings algorithm to sample the posterior distribution but its convergence was very slow so that was why we switched to the Metropolis-within-Gibbs sampler and found that it had much better convergence and stability. Another difference concerns the use of the empirical extreme value distribution and we show that it deviates from the Gumbel distribution. In paper [EJ2017] on page 4 the probability to have an event larger than a specified magnitude is given as 1

– $\exp[-N(m|\text{seq}, M_1)]$. However, it is not clear how this distribution was derived. We do not think that this is the extreme value distribution analogous to Eq. (11) of our revised paper. Therefore, the comparison of our results and the ones derived in [EJ2017] is not that straightforward. But we acknowledge that [EJ2017] paper is using a similar approach to sample the posterior distribution and forecast the evolution of seismicity. In the revised version of the paper we indicate this.

“(2) The paper is of interest to others in the field. Quasi real-time tuning of ETAS models has been a challenge especially in its more general spatio-temporal clustering format. Nevertheless, since the objective of the paper is estimating the maximum magnitude, the time-magnitude version is adopted in the manuscript. The MCMC framework provides the possibility of sampling directly from a conditional distribution (the joint probability distribution of the ETAS model parameters given the events that have already taken place in the sequence) which is known but for a constant. This possibility can be also useful for Bayesian inference of other quantities such as the rate of seismicity.”

Reply:

We agree that we consider only the time dependent version of the ETAS model. We think that for spatially compact sequences this approximation is reasonable. The temporal seismic rate can be computed stochastically by simulating the ETAS process forward in time. In fact, we do such simulations in order to extract the maximum magnitude from each realization of the ETAS process during the forecasting time interval.

“(3) Another point of interest of crucial importance, beyond the technical difficulties associated to parameter estimation, is whether/under which circumstances the ETAS model is able to forecast with a certain confidence events that are larger in magnitude with respect to the main event or to forecast a mainshock based on its foreshocks (if they exist). With regard to this point, the results of forecasting after $t=1.16$ days (shown in the supplementary material Fig. S5) apparently indicate that the forecasts provided for $M>7.3$ right before it took place did not foresee it with a large probability (obviously the interpretation is relative). It would be important to provide some insight with this regard in the discussion of the results”

Reply:

The ETAS model specifically allows to have triggered events, which can be larger in magnitude than the parent events. The actual probabilities to forecast large mainshocks (like the M7.3 Kumamoto mainshock) inferred from the previous seismicity rate are typically rather low. We think that this is limited by the information content the seismicity rate contains. To improve such forecasting, one needs to consider, in addition, other physics-based processes, for example, the stress transfer and more accurate rheological description of the crust and mantle. We added these points into Discussion section. When performing the sensitivity analysis, we also show that the computed probabilities also depend on the mean of the prior distribution of the background rate μ . In the revised version of the paper, we use the point estimates of the ETAS parameters for the mean values of prior distributions and specifically the value of μ as the values for

the mean of the prior distributions for each ETAS parameter. This resulted in slight increase in probabilities for subsequent largest events.

“(4) It would be useful to shed some more insight to the assumptions made for deriving the likelihood in Eq. (4) and how it is implemented. Given that the Bayesian inference based on MCMC is used and given that the prior distributions are assumed to be independent, it is the likelihood function that determines to a large degree the possible correlations between the inter-arrival times and the magnitudes. The formulation in Eq. 4 apparently seems to indicate that the inter-arrival times are expressed as corresponding to a Poisson point process (and therefore independent) and the magnitudes are i.i.d. and also independent from the inter-arrival times. Moreover, the formulation seems also to be based on the assumption that the conditional rate has a functional form that is not going to change during the sequence (non-evolutionary). This somehow seems to be confirmed by the very small difference observed (Fig. S7) between the distribution of maximum magnitude based on Eq. S1 and the non-parametric one based on the samples generated from MCMC-Gibbs sampling.”

Reply:

We would like to clarify this point. The likelihood function given in Eq. (4) is derived under the assumption that the event magnitudes are i.i.d. random variables. Therefore, when one samples the posterior distribution one obtains the Markov sequences for the model parameters. Whereas, the interevent times between events are fully specified by the conditional rate given in Eq. (7). So, the interevent times are not controlled by the likelihood. However, the ETAS model is not a non-homogeneous Poisson process. It is a doubly stochastic process because the rate is not a constant or regularly changing function like the modified Omori law but a stochastic function of time. The observed difference between the distribution for the extreme magnitudes given in Figure 5 (in the revised version of the paper) is also observed when one uses a single set of the ETAS parameters so it is not the result of the MCMC sampling.

“(5) When sampling the ETAS parameters, does the manuscript consider the triggering effect of the events that are going to take place in the forecasting interval? Or the paper assumes that the ETAS rate considers only the triggering of the events taken place before the forecasting interval? This latter assumption is going to simplify the simulation process and would probably affect the results in a non-conservative manner. Please specify in the paper.”

Reply:

We consider the triggering when we simulate the ETAS model during the forecasting time interval. This is important to fully generate the ETAS sequences of events. However, we do not consider the effect of triggered events during the forecasting time interval when sampling the posterior distribution to obtain the distribution of the ETAS parameters. This is done to ensure that we use only the earthquake information during the

training time interval. This also assumes that the parameters of the ETAS model are not changing with time. We clarified this in more details in the revised version of the paper.

“(6) The analytical formulation shown in Section S1.1 is very interesting. It would be curious to see how the results are going to differ from those calculated based on Equation S1. Nevertheless, the paper does not seem to use or demonstrate the implementation/results of the formulations derived in S1.1 --which seem to have a recursive formulation and might be time consuming to calculate. There is a reference to this section in Page 10, line 227 but it was not very clear to the reviewer. Even if the formulation is challenging to be implemented, it would be useful and insightful to mention those challenges.”

Reply:

The obtained integral formulation will reduce significantly the efficiency of the computation of the Bayesian predictive distribution. Because for each MCMC sample of the model parameters one will need to solve it numerically. In addition, due to stochastic nature of the ETAS process the numerical computation of the integrals can be problematic. As a result, the simulation of the EATS model forward in time and the extraction of the maximum magnitude for each simulation is the most efficient and appropriate method here.

“(1) Page 7, line 148: The manuscript adopts the truncated Normal for sampling positive values from the proposal distribution, why Lognormal is not used?”

Reply:

In fact, we implemented the lognormal distribution as well and compared its performance with the truncated normal distribution. They produce statistically the same results. The truncated normal distribution offers more flexibility in controlling the lower and upper bounds of the distribution. This can be helpful when one wants to limit one particular model parameter to a specific value or range. So that is why we use it in the analysis. In the revised version of the paper we added Figure S7, where we compare several combinations of the proposal and prior distribution.

“(2) The two algorithms reported in the Methods sections were not very easy to follow for the reviewer. Please either explain in more detail the notation used or replace them with less detailed and more conceptual diagrams.”

Reply:

We expanded the description of the algorithms to clarify them in more detail.

“(3) Page 3, Line 74: What is the upper bound for i ?”

Reply:

We did not provide the upper bound in this general description as the total number of events in a finite time interval depends on the lower magnitude cutoff. Later in the text, where we define the posterior distribution or other quantities, we specify explicitly the number of events used.

“(4) Is Equation S2 providing the numerical Monte Carlo estimate of Eq 1? If this is the case, please specify. The paragraph following Eq. S2 was not very clear with this regard.”

Reply:

Yes, that is the case. We added text to clarify this.

“(5) Page 3 supplementary material: by complete probability it is intended “total probability”?”

Reply:

We corrected this.

“(6) Figure S9c what are the transformed time and earthquake index? Please explain.”

Reply:

We added the explanation.

Reply to the review of the second referee:

We would like to thank the reviewer for his/her constructive comments and suggestions. They were appropriate, and we tried to address them in order to clarify the analysis and justify the results.

“In my opinion, the authors could improve the manuscript primarily with more descriptive language, and by using more direct comparisons between this method and past practice (I did find this in the supplements, but I think it should be in the main body of the paper).”

Reply:

In the revised version of the paper we moved several sections from the supplementary materials into the main text. We also improved the overall text and presentation to make more accessible.

“(1) “...an earthquake sequence governed by the ETAS process” I think it is important for a general audience to fully describe ETAS, and clarify that it is a statistical method that does not govern, but attempts to describe physics through empiricism.”

Reply:

We added more extended explanation of the ETAS model.

“(2) Early in the draft it is stated: “In addition, we show that the extreme value distribution for the magnitudes of the largest events in the ETAS model deviates from the Gumbel distribution indicating a certain degree of correlation between event magnitudes”

This is an important result that shows why your work is important, and people may not grasp this without a definition of the Gumbel distribution. I'd suggest a couple more lines here using plain language.”

Reply:

We added the definition of the Gumbel distribution into the text.

“(3) “For the prior distribution, $p(q;w)$, of the model parameters, we used a Gamma distribution (specific values for the mean and variance of the priors for each model parameter are provided in Table S1). For the proposal distribution $J(x|x^{\sim})$, we used a truncated normal distribution ...”

I'm sure these are the right choices, but I think it detracts a little from the paper when certain distributions/parameters are used without stating why, nor what the impact would be to use different choices. That said, for the most part the sensitivity analyses are commendable. I'm just suggesting some additional text in these cases”

Reply:

For the proposal distribution, in addition, we also implemented the lognormal distribution. Both the truncated normal distribution and the lognormal distribution produce statistically the same results. We added Figure S7 to the Supplementary Material illustrating this. In the literature on MCMC methods there is no a prescribed recipe for the choice of a proposal distribution. So, it is more of a trial and error approach. Typically, the proposal distribution needs to be close to the distribution which is sampled and the normal distribution is frequently used. In our case, we considered the truncated normal distribution in order to have positive parameters and have better control on the bounds of those parameters. But we also implemented the lognormal distribution and now state this in the paper.

For the prior distribution, the gamma distribution was a natural choice as it is defined to have positive values. We also tested the truncated normal distribution. They both produce comparable results. We added one extra figure (Figure S7) in Supplementary Material where the compare several possible combinations of the proposal distribution and the prior distributions. We added the corresponding explanation into the revised text.

“(4) Results: “For a fixed probability level of 5% the estimated interarrival times are $DT = 0:26$, $0:81$, and $1:38$ days, respectively. For a 10% probability level the estimated interarrival times are $DT = 0:56$, $2:26$, and $4:52$ days.”

It's probably worthwhile to place results like these into forecast context, and in so doing maybe drop the significant figures back a little.”

Reply:

We reduced significant figures for all estimated probabilities and time intervals.

“Last point: I wonder how this method would work in a region where there are very large faults. Would the exponential magnitude-frequency distribution require modification to account for background probability of very high magnitudes on these structures that might originate from outside of the region where the ETAS parameters apply?”

Reply:

In the suggested framework we adopt a standard model for the frequency-magnitude statistics for earthquakes, i.e. the most commonly used left-truncated exponential distribution. One possible way to include the presence of large faults that can generate strong earthquakes is to consider a characteristic earthquake hypothesis, where there is an elevated probability of having large earthquakes compared to the exponential distribution. This can be done similarly as in Youngs and Coppersmith (BSSA, v.75, 1985, p. 939). This will introduce additional parameters into the problem, but the developed framework allows to consider more complicated models for the frequency-magnitude statistics and earthquake rates. So, this can be implemented in the future studies.

Reply to the review of the third referee:

We would like to thank the reviewer for his/her constructive comments and criticism.

“However, I fail to see much novelty. It mostly represents some incremental research on topics and methods already well established:

1) Dozens of articles if not hundreds use ETAS as baseline model.

2) Bayesian approaches to estimate M_{max} are already available in the literature, as already stated by the authors.”

Reply:

We agree that many works use the ETAS model. But in the current work, we, for the first time, incorporated the full time-dependent ETAS model into the Bayesian predictive framework. This framework assumes the full incorporation of the uncertainties of the model parameters when computing various probabilities etc. We also show that the ETAS process generates maximum magnitude events and their distribution deviates from the standard Gumbel distribution.

We would like to clarify that we are not estimating M_{max} here. M_{max} is related to the estimation of the maximum possible magnitude in a particular seismogenic zone. Here we address the problem of constraining the magnitude of the expected largest earthquake to occur in a finite time interval governed by a specific stochastic point process for a given sequence of earthquakes.

“Although I'm not certain that Bayes' Theorem has not already been applied for ETAS, the new implementation will mostly interest statisticians. How this new approach performs compared to non-epidemic type models has yet to be proven and this will require model comparisons for numerous earthquake sequences. Here the study case is only illustrative in nature. Moreover, in operational earthquake forecasting (OEF), first responders can already use simple Omori statistics (or variants such as STEP) to estimate large aftershock probabilities. They can then reiterate to model subsequent aftershocks. In view of the large number of parameters required for ETAS, I'm not so sure that the added uncertainties will help improve OEF..”

Reply:

In our work, we use several aspects of the Bayesian analysis. Bayes' theorem is used to construct the posterior distribution of the model parameters. We implemented a robust approach to sample this posterior distribution and estimate the variability of the ETAS parameters using the Markov Chain Monte Carlo sampling. We computed for the first time the extreme value distribution for the ETAS model and showed that it deviates from the Gumbel distribution. Then we consider the less used aspect of Bayesian analysis, i.e.

the construction of the Bayesian predictive distribution for the magnitude of the largest expected earthquake. This allows to fully incorporate the uncertainties of the model parameters into the computation of the relevant probabilities.

Most traditional approaches use point estimates for their respective model parameters. So they do not fully take into account the uncertainties associated with model parameters or they partially do by certain bootstrapping methods. In our proposed and implemented method we address this. We also show the comparison with other methods, specifically if one uses the modified Omori law. In our previous work (Shcherbakov et al., GJI 2018) we compared the Bayesian predictive approach with a more traditional approach based on Reasenberg and Jones (1989) model to compute the magnitudes of expected largest events assuming the modified Omori law for the aftershock rate. In the revised version, in addition, we examined several more past aftershock sequences and compared the obtained results.

The proposed approach can incorporate more advanced stochastic models describing the occurrence of earthquakes. So, it has the ability to go beyond the ETAS model.

“Finally, when discussing about "extreme earthquakes", the authors should be more cautious. ETAS, in this context, is misleading since it represents a point process while large earthquakes occur on existing faults, whose geometry and loading state are not included in ETAS. Therefore, the model is unlikely to be representative of the ongoing physical processes in the crust.”

Reply:

We agree with the reviewer that the ETAS model is only an approximation to real earthquake processes. The inclusion of large existing faults can be accomplished by specifying a more elaborate frequency-magnitude distribution, for example, assuming a characteristic earthquake hypothesis. This can be done similarly as in Youngs and Coppersmith (BSSA, v.75, 1985, p. 939). When such a model is formulated it can be incorporated into our suggested framework. We think this is the power of Bayesian analysis that any prior information can be used to improve the forecasts. Please also see our reply to the last point of Reviewer 2.

Reviewers' comments:

Reviewer #1 (Remarks to the Author):

The authors have responded in a satisfactory manner to most of the points raised by the reviewer in the previous round. Here are a couple of observations regarding the revised paper:

- 1) It seems somehow inconsistent to present in the methods section a formulation for the probability of exceeding a certain magnitude threshold for ETAS model (under the title: The extreme value distributions for the ETAS process). However, this methodology is not used nor compared in the paper with the distribution obtained empirically and through simulation of the sequence of events. This would have been a perfect way for validating the results.
- 2) The presented method seems to offer little or no improvement in terms of its predictive capability of events with large magnitudes with respect to simpler models such as Gumble extreme value based on the Omori-Utsu law.
- 3) The fact that the authors do not consider the triggered events in the forecasting interval might lead to underestimation of the probability of exceeding certain magnitude thresholds.
- 4) The authors emphasize the statistically significant difference between the Gumble distribution and the ETAS-based empirically-estimated extreme value distribution. However, to the eye, this seems like a small difference (Figure S8).
- 5) In response to the point raised by the reviewer regarding the likelihood function, the authors state that indeed the formula is based on assumption of independent inter-arrival times and independent magnitudes. The authors state that the correlation between the inter-arrival times is embedded in the functional form of the ETAS rate. However, in reviewer's opinion, the correlation between the inter-arrival times needs to be implemented formally as a conditional probability and not just embedded in the functional form of the rate (at the very least when the purpose is simulating the events and their inter-arrival times in order to empirically calculate the extreme value distribution for ETAS). At the very least, the authors are invited to provide more details on how the sequences (inter-arrival times) are sampled in the forecasting interval.

Fatemeh Jalayer
28/05/2019

Reviewer #2 (Remarks to the Author):

I think that the authors have responded to my comments sufficiently. I have no further issues.

Reviewer #3 (Remarks to the Author):

The authors have done a diligent work at proving the added value of their approach. I have no more concern.

Reply to the second review of the first referee:

We would like to thank Dr. Jalayer for her criticism and help to improve the justification of the results. They helped to enhance the presentation and clarify several aspects of the analysis. Please, find below our response to several additional points raised by the reviewer.

“It seems somehow inconsistent to present in the methods section a formulation for the probability of exceeding a certain magnitude threshold for ETAS model (under the title: The extreme value distributions for the ETAS process). However, this methodology is not used nor compared in the paper with the distribution obtained empirically and through simulation of the sequence of events. This would have been a perfect way for validating the results.”

Reply:

In the first round of reviews, there were diverging opinions of two reviewers: one suggested to move the analytical calculations from Supplementary Information into the main body (as we did it), the other reviewer wanted to have the main paper less technical and more illustrative. We agree that the analytically derived integral equation for the extreme value distribution is not used when computing the predictive distribution. Therefore, in the revised version of the paper, we moved it back to Supplementary Information. We think that it is a new result and it is important to have it in the paper. The proper numerical treatment of the integral equation is a complicated task, which is beyond the scope of this paper. In future work, it will be interesting to consider this exercise and to compare with the stochastic simulation of the ETAS model.

“The presented method seems to offer little or no improvement in terms of its predictive capability of events with large magnitudes with respect to simpler models such as Gumble extreme value based on the Omori-Utsu law.”

Reply:

We disagree with the reviewer. We demonstrate that, when considering complex earthquake sequences such as the 2016 Kumamoto sequence, modeling the earthquake rate using the ETAS process makes a substantial difference. This is clearly illustrated in Figure 5, where inclusion of foreshocks produces higher probabilities for the largest expected aftershocks of the mainshock (solid brown curve vs. solid pink curve). For the other several aftershock sequences (Table 2), we considered only aftershocks of each mainshock. It is not surprising that the Omori-Utsu law dominates the aftershock probabilities for those large mainshocks, and that the use of ETAS provides higher-order corrections. These corrections are, however, important for the practical estimation of the hazard related to rare events, e.g. for Kumamoto, where $M_{ms} - m_{as}$ is only 1.5, the difference between ETAS and OU (OU_MCMC) makes 9% (5%). Large events have always low probabilities of occurrence and are associated with high uncertainties. Therefore, additional information that allows to improve the reliability of forecasts, should be taken into account. In addition, the suggested method goes beyond typical mainshock-aftershock sequences. It can be used for any type of seismicity, where

clustering and triggering of earthquakes is present. This clearly illustrates the advantage of the proposed method compared to the standard assumption of the Omori-Utsu law combined with the Gumbel distribution, which is only applicable to well-approximated by the Omori-Utsu formula aftershock sequences.

“The fact that the authors do not consider the triggered events in the forecasting interval might lead to underestimation of the probability of exceeding certain magnitude thresholds.”

Reply:

This seems to be a misunderstanding. In our work, we fully consider triggered events during the forecasting time interval by simulating the ETAS process. We performed the simulation of the ETAS process using the well-established thinning algorithm. This is implemented in the Matlab file `etas_gen.m` and was provided as a part of the last resubmission. To clarify this, we added a statement to the revised text about how the ETAS process was simulated in the forecasting time interval.

“The authors emphasize the statistically significant difference between the Gumble distribution and the ETAS-based empirically-estimated extreme value distribution. However, to the eye, this seems like a small difference (Figure S8).”

Reply:

We added the results of the Kolmogorov-Smirnov test to show that they are statistically different. The KS test applied to the two distributions given in Figure S8 showed that the null hypothesis that they came from the same distribution was rejected at the 1% significance level.

“In response to the point raised by the reviewer regarding the likelihood function, the authors state that indeed the formula is based on assumption of independent inter-arrival times and independent magnitudes. The authors state that the correlation between the inter-arrival times is embedded in the functional form of the ETAS rate. However, in reviewer’s opinion, the correlation between the inter-arrival times needs to be implemented formally as a conditional probability and not just embedded in the functional form of the rate (at the very least when the purpose is simulating the events and their inter-arrival times in order to empirically calculate the extreme value distribution for ETAS). At the very least, the authors are invited to provide more details on how the sequences (inter-arrival times) are sampled in the forecasting interval.”

Reply:

I think we need to clarify this further. The likelihood function for Hawkes type of processes (the ETAS process is a particular example) is well studied mathematically and details can be found, for example, in Daley D.J. and Vere-Jones D., *An Introduction to the Theory of Point Processes*, Volume 1, Springer, 2003. We use this well-established form of the likelihood function for the ETAS process in our analysis. We do not think that we should deviate from this form. When constructing the likelihood function for the ETAS process, Eq. (4), it is assumed that the magnitudes are i.i.d. But it does not assume that inter-event times are uncorrelated or i.i.d. In fact, the inter-event times $t_i - t_{i-1}$ are correlated for the Omori-Utsu kernel, as after each subsequent event, the time interval to

the next event is going to be, on average, longer. In addition, the time interval to the next event is dependent on the current magnitude m_i , i.e. large magnitude events lead to small interevent times to the next event and this is taken into account in the product of the rates $\lambda(t_i)$ in Eq. (4). This example illustrates that in case of nonstationary processes one can observe correlations between inter-event times. The notion of i.i.d. and non-i.i.d. is reserved for random variables and the magnitudes are such an example and we assume that they are i.i.d. However, this is not true for interevent times. In the revised version, we clarified this further. In our work, we simulate the ETAS process using the well-established thinning algorithm. In this algorithm the time interval to the next event is generated based on a Poisson process but accepted with a certain probability depending on the conditional rate of the EATS process. This is implemented in the Matlab file `etas_gen.m` and was provided as a part of the last resubmission. We added a statement to the revised text about how the ETAS process was simulated in the forecasting time interval.

Reply to the review of the second referee:

“I think that the authors have responded to my comments sufficiently. I have no further issues.”

Reply:

We would like to thank the reviewer for his/her constructive comments and suggestions. They helped to improve the analysis and overall presentation of the results.

Reply to the review of the third referee:

“The authors have done a diligent work at proving the added value of their approach. I have no more concern.”

Reply:

We would like to thank the reviewer for his/her comments and criticism that improved and clarified the results in the paper.

REVIEWERS' COMMENTS:

Reviewer #1 (Remarks to the Author):

The authors have responded to all the points I had raised in the previous review(s). I have no further issues.